# Falconn++: A Locality-sensitive Filtering Approach for Approximate Nearest Neighbor Search

**Ninh Pham**
School of Computer Science
University of Auckland
ninh.pham@auckland.ac.nz

**Tao Liu**
School of Computer Science
University of Auckland
tliu137@aucklanduni.ac.nz

## Abstract

We present Falconn++, a novel locality-sensitive filtering approach for approximate nearest neighbor search on angular distance. Falconn++ can filter out potential far away points in any hash bucket *before* querying, which results in higher quality candidates compared to other hashing-based solutions. Theoretically, Falconn++ asymptotically achieves lower query time complexity than Falconn, an optimal locality-sensitive hashing scheme on angular distance. Empirically, Falconn++ achieves higher recall-speed tradeoffs than Falconn on many real-world data sets. Falconn++ is also competitive with HNSW, an efficient representative of graph-based solutions on high search recall regimes.

## 1 Introduction

Nearest neighbor search (NNS) on a unit sphere is the task of, given a point set $\mathbf{X} \subset \mathcal{S}^{d-1}$ of size $n$ and a query point $\mathbf{q} \in \mathcal{S}^{d-1}$, finding the point $\mathbf{p} \in \mathbf{X}$ such that $\mathbf{p} = \arg\min_{\mathbf{x} \in \mathbf{X}} \|\mathbf{x} - \mathbf{q}\|$. Since both data set $\mathbf{X}$ and query $\mathbf{q}$ are on a unit sphere, NNS on Euclidean distance is identical to NNS on angular distance (i.e. $\mathbf{p} = \arg\max_{\mathbf{x} \in \mathbf{X}} \mathbf{x}^\top \mathbf{q}$). NNS and its variant top-$k$ NNS on angular distance are the fundamental problems in computer science, e.g. web search [11], recommender system [21], machine learning [7], and computer vision [22].

NNS dates back to Minsky and Papert [17], who studied the tradeoff between query time and indexing space in high dimensions. Due to the "curse of dimensionality", linear scan outperforms space/data partitioning solutions on solving exact NNS given a linear indexing space [24]. Given a polynomial indexing space $n^{\mathcal{O}(1)} d^{\mathcal{O}(1)}$, there is no known algorithm to solve exact NNS in truly sublinear time. If there exists such an algorithm, the Strong Exponential Time Hypothesis (SETH) [12], a fundamental conjecture in computational complexity theory, is wrong [3, 25].

Due to the hardness of exact NNS, researchers study efficient solutions for approximate NNS (ANNS). Given an approximation factor $c' > 1$, ANNS aims at finding the point $\mathbf{p}' \in \mathbf{X}$ within distance $c' \cdot \|\mathbf{p} - \mathbf{q}\|$. Since we can reduce ANNS to its "decision version", called $(c, r)$-approximate near neighbor ($(c, r)$-NN) for an approximation factor $c > 1$ and a radius $r$ [10], solving ANNS boils down to $(c, r)$-NN. The problem $(c, r)$-NN that asks to return any point within distance $cr$ to $\mathbf{q}$ if there is a point within distance $r$ to $\mathbf{q}$ is the key algorithmic component of many ANNS solvers. *Locality-sensitive hashing* (LSH) [13] is the first *provable* sublinear query time solution using a subquadratic indexing space for $(c, r)$-NN in high dimensions.

In principle, LSH hashes "close" points, i.e. within radius $r$ to $\mathbf{q}$, into the same bucket with $\mathbf{q}$ with high probability, and hashes "far away" points, i.e. outside radius $cr$ to $\mathbf{q}$, into the same bucket with low probability. Given a parameter $0 < \rho < 1$, LSH answers $(c, r)$-NN in $\mathcal{O}(dn^\rho)$ time and uses $\mathcal{O}(nd + n^{1+\rho})$ space. Since the exponent $\rho$ governs the complexity of LSH, it is desirable to have LSH families with the smallest $\rho$. LSH families with optimal values $\rho \approx 1/c^2$ on Euclidean

distance are well studied in theory [1, 5]. In practice, to our knowledge, only Falconn [2] with the cross-polytope LSH family asymptotically achieves $\rho \approx 1/c^2$ on the Euclidean unit sphere.

Despite provable sublinear query time guarantees, the indexing space of LSH is a primary bottleneck. Since different queries require different values of $r$, practical LSH-based solutions often use hundreds of hash tables to guarantee the querying performance. To overcome this bottleneck, Lv et al. [15] propose a multi-probing trick that sequentially checks nearby buckets of the query's bucket. This is because the hash values of the nearest neighbors of $\mathbf{q}$ tend to be close to the hash value of $\mathbf{q}$ on the hashing space [19]. Therefore, multi-probe LSH achieves a similar success probability with a smaller number of hash tables than the standard LSH.

We observe that LSH and its multi-probe schemes do not efficiently answer ANNS on high recall regimes. Since the distance gaps between the nearest neighbors are tiny in many real-world data sets, we need a very small approximation factor $c$, e.g. $c = 1.01$. This requirement degrades the performance of LSH-based solutions as $\rho \to 1$. In this case, multi-probe LSH has to increase the number of probes and the number of candidates. That significantly reduces the querying performance since the probed buckets different from the query bucket tend to have more far away points. Empirically, multi-probe Falconn [2] with $L = 100$ hash tables on the Glove300 data set needs 4,000 probes and computes more than $0.1 \cdot n$ number of distances to achieve 90% recall. This is the rationale that LSH-based solutions are less favorable for ANNS on high recall regimes than the recent graph-based solutions [6], e.g. HNSW [16].

To further facilitate LSH-based solutions for ANNS, we propose a *filtering mechanism* that can efficiently filter out the far away points in any bucket *before* querying. Our proposed mechanism is also *locality-sensitive* in the sense that far away points will be filtered out with high probability and close points will be pruned with low probability. Therefore, we can safely remove far away points while building the index to improve the query time and reduce the indexing space simultaneously. This feature significantly increases the performance of multi-probe LSH since it dramatically reduces the number of distance computations while preserving the search recall.

We implement the proposed filtering mechanism for angular distance based on the asymptotic property of the concomitant of extreme order statistics [9, 20]. Similar to Falconn [2], our approach utilizes many random projection vectors and only considers specific vectors $\mathbf{r}_i$ that are closest or furthest to the query $\mathbf{q}$. The key difference is that we use the random projection value $\mathbf{x}^\top \mathbf{r}_i$ to estimate $\mathbf{x}^\top \mathbf{q}$ by leveraging the asymptotic behavior of these projection values. This feature makes our filtering mechanism *locality-sensitive*.

Theoretically, our proposed locality-sensitive filtering (LSF) scheme, named *Falconn++*, asymptotically achieves smaller $\rho$, a parameter governing the query time complexity, than Falconn. Empirically, Falconn++ with a new add-on that applies multi-probing on *both* querying and indexing phases achieves significantly higher recall-speed tradeoffs than Falconn on many real-world data sets. We further innovate Falconn++ with several heuristics that make it competitive with HNSW [16], a recent efficient graph-based solution for ANNS on high recall regimes.

## 2  Preliminary

### 2.1  Locality-sensitive hashing (LSH)

Since 1998, LSH [13] has emerged as an algorithmic primitive of ANNS in high dimensions. LSH *provably* guarantees a sublinear query time for ANNS by solving the $(c, r)$-NN over $\tilde{\mathcal{O}}(\log n)$ different radii. Given a distance function $dist(\cdot, \cdot)$, the LSH definition is as follows.

**Definition 1** (LSH). *For positive reals $r$, $c$, $p_1$, $p_2$, and $0 < p_2 < p_1 \leq 1$, $c > 1$, a family of functions $\mathcal{H}$ is $(r, cr, p_1, p_2)$-sensitive if for uniformly chosen $h \in \mathcal{H}$ and all $\mathbf{x}, \mathbf{y}, \mathbf{q} \in \mathbb{R}^d$:*

- *If $dist(\mathbf{x}, \mathbf{q}) \leq r$, then $\mathbf{Pr}\left[h(\mathbf{x}) = h(\mathbf{q})\right] \geq p_1$ ;*
- *If $dist(\mathbf{y}, \mathbf{q}) \geq cr$, then $\mathbf{Pr}\left[h(\mathbf{y}) = h(\mathbf{q})\right] \leq p_2$ .*

To simplify notations, we call $\mathbf{x}$ as a "close" point if $dist(\mathbf{x}, \mathbf{q}) \leq r$ and $\mathbf{y}$ as "far away" point if $dist(\mathbf{y}, \mathbf{q}) \geq cr$. To minimize the collision probability of far away points, we combine $l = \mathcal{O}(\log n)$ different LSH functions together to form a new $(r, cr, p_1^l, p_2^l)$-sensitive LSH family, and set $p_2^l = 1/n$. Since the combination also decreases the collision probability of close points from $p_1$ to $p_1^l$, we need

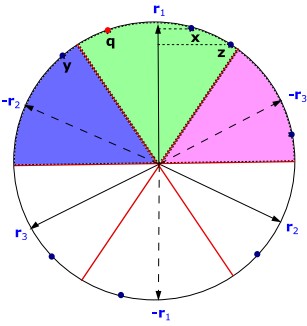

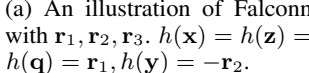

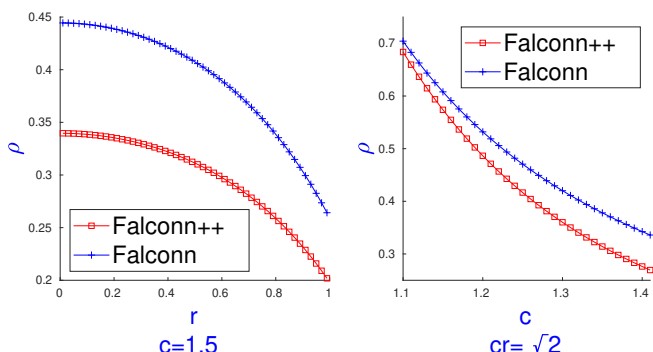

(a) An illustration of Falconn with $\mathbf{r}_1, \mathbf{r}_2, \mathbf{r}_3$. $h(\mathbf{x}) = h(\mathbf{z}) = h(\mathbf{q}) = \mathbf{r}_1, h(\mathbf{y}) = -\mathbf{r}_2$.

(b) A comparison of $\rho$ between Falconn++ and Falconn while fixing $c = 1.5$ and varying $r$; and fixing $cr = \sqrt{2}$ and varying $c$.

$L = \mathcal{O}\left(1/p_1^l\right)$ hash tables to ensure that we can find a close point with a constant probability. Given the setting $p_2^l = 1/n$ and denote by $\rho = \ln(1/p_1)/\ln(1/p_2)$, LSH needs $L = \mathcal{O}\left(n^\rho\right)$ tables and compute $\mathcal{O}\left(n^\rho\right)$ distances in $\mathcal{O}\left(dn^\rho\right)$ time to answer $(c, r)$-NN with a constant probability.

The exponent $\rho$ governing the complexity of an $(r, cr, p_1, p_2)$-sensitive LSH algorithm depends on the distance function and $c$. O'Donnell et al. [18] show that $\rho \geq 1/c^2 - o(1)$ on Euclidean distance given $p_2 \geq 1/n$. However, it is difficult to implement the LSH families with optimal $\rho$ in practice.

## 2.2 Falconn: An optimal LSH on angular distance

Andoni et al. [2] introduce *Falconn* that uses a cross-polytope LSH family on angular distance (i.e. cosine similarity). Given $D$ random vectors $\mathbf{r}_i \in \mathbb{R}^d, i \in [D]$ whose coordinates are randomly selected from the standard normal distribution $N(0, 1)$, and $\mathtt{sgn}(\cdot)$ is the sign function, the cross-polytope LSH family uses the index of the random vector as the hash value. In particular, w.l.o.g. we assume that $\mathbf{r}_1 = \arg\max_{\mathbf{r}_i} |\mathbf{q}^\top \mathbf{r}_i|$, then $h(\mathbf{q})$ is calculated as follows:

$$h(\mathbf{q}) = \begin{cases} \mathbf{r}_1 \text{ if } \mathtt{sgn}(\mathbf{q}^\top \mathbf{r}_1) \geq 0, \\ -\mathbf{r}_1 \text{ otherwise}. \end{cases} \tag{1}$$

A geometric intuition of Falconn is that $h(\mathbf{q}) = \mathbf{r}_1$ (or $h(\mathbf{q}) = -\mathbf{r}_1$) indicates that $\mathbf{q}$ is closest (or furthest) to the random vector $\mathbf{r}_1$ among $D$ random vectors, as shown in Figure 1a. Falconn uses $D$ random vectors $\mathbf{r}_i$ to partition the sphere into $2D$ wedges centered by the $2D$ random vectors $\pm \mathbf{r}_i$. If $\mathbf{x}$ and $\mathbf{q}$ are similar, they tend to be closest or furthest to the same random vector, and hence are hashed into the same wedge corresponding to that random vector. The following lemma states the asymptotic collision probability between two points $\mathbf{x}, \mathbf{q}$ when $D$ is sufficiently large.

**Lemma 1.** *[2, Theorem 1] Given two points $\mathbf{x}, \mathbf{q} \in \mathcal{S}^{d-1}$ such that $\|\mathbf{x} - \mathbf{q}\| = r$ where $0 < r < 2$. Then, we have*

$$\mathbf{Pr}\left[h(\mathbf{x}) = h(\mathbf{q})\right] \approx D^{-\frac{1}{4/r^2-1}}. \tag{2}$$

Given a small $c$, Equation 2 indicates that Falconn has an optimal exponent $\rho \approx \frac{4/c^2r^2-1}{4/r^2-1} \approx 1/c^2$.

**Multi-probe Falconn.** To reduce the number of hash tables $L$, multi-probe Falconn considers the next closest or furthest random vectors to generate the probing sequence in a hash table. For example, in Figure 1a, since $\mathbf{q}$ is next closest to $-\mathbf{r}_2$, the bucket corresponding to the blue wedge will be the next probe. Given a fixed total number of query probes $qProbes$ for a query, ranking the buckets across $L$ tables is challenging. Falconn heuristically uses the distance between the projection values of $\mathbf{q}$ on random vectors to construct the probing sequence. For example, in the table corresponding to Figure 1a, the ranking score of the blue wedge for probing is $|\mathbf{q}^\top \mathbf{r}_1| - |\mathbf{q}^\top \mathbf{r}_2|$. Even though multi-probe Falconn can significantly reduce the number of hash tables, it demands a significantly large $qProbes$ and candidate sizes to solve ANN on high recall regimes due to the small distance gaps among nearest neighbors.

## 2.3 CEOs: An optimal dimensionality reduction on unit sphere

Pham proposes CEOs [20], an optimal dimensionality reduction for ANN on angular distance. While CEOs shares the same spirit that utilizes random projections as Falconn, it is a similarity estimation technique rather than hashing. In particular, CEOs randomly projects $\mathbf{X}$ and $\mathbf{q}$ onto $D$ Gaussian random vectors and considers the projection values on specific random vectors that are closest or furthest to $\mathbf{q}$, e.g. $\arg\max_{\mathbf{r}_i} |\mathbf{q}^\top \mathbf{r}_i|$. Asymptotically, given a sufficiently large $D$, the following lemma states the projection values on a few specific random vectors preserve the inner product.

**Lemma 2.** *[20] Given two points $\mathbf{x}, \mathbf{q} \in \mathcal{S}^{d-1}$ and significantly large $D$ random vectors $\mathbf{r}_i$, w.l.o.g we assume that $\mathbf{r}_1 = \arg\max_{\mathbf{r}_i} |\mathbf{q}^\top \mathbf{r}_i|$. Then, we have*

$$\mathbf{x}^\top \mathbf{r}_1 \sim N\left(\text{sgn}(\mathbf{q}^\top \mathbf{r}_1) \cdot \mathbf{x}^\top \mathbf{q}\, \sqrt{2\ln D}, 1 - (\mathbf{x}^\top \mathbf{q})^2\right). \tag{3}$$

Among $D$ random vectors, if $\mathbf{r}_1$ is closest or furthest to $\mathbf{q}$, the projection values of all points in $\mathbf{X}$ preserve their inner product order in expectation. In Figure 1a, since $\mathbf{q}$ is closest to $\mathbf{r}_1$, the order of projection values onto $\mathbf{r}_1$, e.g. $\mathbf{x}^\top \mathbf{r}_1 > \mathbf{z}^\top \mathbf{r}_1$, preserves the inner product order, e.g. $\mathbf{x}^\top \mathbf{q} > \mathbf{z}^\top \mathbf{q}$.

Importantly, given a constant $s_0 > 0$, Lemma 2 also holds for the top-$s_0$ closest vectors and top-$s_0$ furthest random vectors to $\mathbf{q}$ due to the asymptotic property of extreme normal order statistics [9]. For example, in Figure 1a, we can see that the inner product order is also preserved on $-\mathbf{r}_1, -\mathbf{r}_2$.

**Connection to Falconn.** From Equations 1 and 3, we can see that Falconn uses the top-1 closest or furthest random vector (i.e. $\mathbf{r}_1$) as the hash value of $\mathbf{q}$ while CEOs uses the projection values over this random vector as inner product estimates. Furthermore, the multi-probe Falconn likely probes buckets corresponding to the $2s_0$ random vectors closest or furthest to $\mathbf{q}$. Since the inner product order is well preserved on the projections onto these $2s_0$ vectors, thanks to the CEOs property, we can keep a small fraction of number of points with the largest projection values in Falconn's buckets. This filtering approach will significantly save the indexing space and improve the query time while maintaining the same recall as multi-probe Falconn. The next section will theoretically analyze this approach and describe its practical implementation.

# 3 Falconn++: A locality-sensitive filtering approach

## 3.1 A locality-sensitive filtering (LSF)

Let $0 < q_2 < q_1 \leq 1$, we define an *LSF mechanism* that can keep in any bucket $h(\mathbf{q})$ any close point $\mathbf{x}$ with probability at least $q_1$ and any far away point $\mathbf{y}$ with probability at most $q_2$ as follows.

**Definition 2** (LSF). *For positive reals $r$, $c$, $q_1$, $q_2$, and $0 < q_2 < q_1 \leq 1$, $c > 1$, given $\mathbf{x}, \mathbf{y}$ in the bucket $h(\mathbf{q})$ where $h$ is randomly chosen from a LSH family, an LSF mechanism is a filter that have the following properties:*

- *If $dist(\mathbf{x}, \mathbf{q}) \leq r$, then $\mathbf{Pr}\left[\mathbf{x} \text{ is not filtered}\right] \geq q_1$ ;*
- *If $dist(\mathbf{y}, \mathbf{q}) \geq cr$, then $\mathbf{Pr}\left[\mathbf{y} \text{ is not filtered}\right] \leq q_2$ .*

Applying the LSF mechanism on each bucket of the hash table constructed by any $(r, cr, p_1, p_2)$-sensitive LSH, the new collision probabilities among $\mathbf{x}, \mathbf{y}, \mathbf{q}$ after hashing and filtering are:

$$\mathbf{Pr}\left[h(\mathbf{x}) = h(\mathbf{q}), \mathbf{x} \text{ is not filtered}\right] \geq q_1 p_1, \ \mathbf{Pr}\left[h(\mathbf{y}) = h(\mathbf{q}), \mathbf{y} \text{ is not filtered}\right] \leq q_2 p_2.$$

Assume that LSF has the property $\ln(1/q_1)/\ln(1/q_2) \leq \rho$, applying hashing and filtering, we can solve $(c, r)$-NN with the new exponent $\rho' \leq \rho$.

**Benefits of LSF.** Given the condition $\ln(1/q_1)/\ln(1/q_2) \leq \rho$ of LSF, combining filtering and hashing mechanisms leads to a faster LSH-based $(c, r)$-NN solver since $\rho' \leq \rho$. We note that the number of required tables is now $\mathcal{O}(1/q_1 p_1)$ whereas the standard LSH only needs $\mathcal{O}(1/p_1)$. Since the number of close points is often much smaller than the number of far away points in real-world data sets, an LSF table contains substantially less than $nq_1$ points in practice. Therefore, the combination of LSF and LSH improves *both* query time and indexing space compared to LSH.

**Challenges.** The challenge is how to design an LSF mechanism with $\ln(1/q_1)/\ln(1/q_2) \leq 1/c^2$ since Falconn has an asymptotically optimal $\rho \approx 1/c^2$. If this condition does not hold, the LSF

mechanism will increase the exponent $\rho$ and degrade the querying performance. For example, randomly discarding a point in a bucket will increase $\rho$. Next, we will instantiate the LSF mechanism based on the CEOs property, which leads to an evolution of Falconn for angular distance.

## 3.2 LSF with CEOs

Given sufficiently large $D$ random vectors $\mathbf{r}_i$, w.l.o.g. we assume $h(\mathbf{q}) = \mathbf{r}_1 = \arg\max_{\mathbf{r}_i} \mathbf{q}^\top \mathbf{r}_i$. We denote by random variables $X = \mathbf{x}^\top \mathbf{r}_1, Y = \mathbf{y}^\top \mathbf{r}_1$ corresponding to $\mathbf{x}, \mathbf{y}$ in the bucket $h(\mathbf{q})$. From Lemma 2 we have $X \sim N\left(\mathbf{x}^\top\mathbf{q}\sqrt{2\ln D}, 1 - \left(\mathbf{x}^\top\mathbf{q}\right)^2\right), Y \sim N\left(\mathbf{y}^\top\mathbf{q}\sqrt{2\ln D}, 1 - \left(\mathbf{y}^\top\mathbf{q}\right)^2\right)$. Exploiting this property, we propose the filtering mechanism as follows:

*Given a threshold $t = (1 - r^2/2)\sqrt{2\ln D}$, in each bucket, we keep any point $\mathbf{x}$ if $\mathbf{x}^\top \mathbf{r}_1 \geq t$ and discard any point $\mathbf{y}$ if $\mathbf{y}^\top \mathbf{r}_1 < t$.*

The following theorem shows that our mechanism is locality-sensitive.

**Theorem 1.** *Given sufficiently large $D$ random vectors and $c > 1$, the proposed filtering mechanism asymptotically has the following properties:*

- *If $\|\mathbf{x} - \mathbf{q}\| \leq r$, then $\mathbf{Pr}\left[\mathbf{x} \text{ is not filtered}\right] \geq q_1 = 1/2$;*
- *If $\|\mathbf{y} - \mathbf{q}\| \geq cr$, then $\mathbf{Pr}\left[\mathbf{y} \text{ is not filtered}\right] \leq q_2 = \frac{1}{\gamma\sqrt{2\pi}} \exp(-\gamma^2/2) < q_1$ where*

$$\gamma = \frac{cr(1-1/c^2)}{\sqrt{4-c^2r^2}} \cdot \sqrt{2\ln D}.$$

The proof uses a classical tail bound on normal variables $X, Y$ over $t$ (see the supplementary materials for details). Given a sufficiently large $D$, we can make $\ln(1/q_1)/\ln(1/q_2)$ arbitrarily smaller than $1/c^2$. Hence, the proposed LSF mechanism can be used to improve Falconn for ANNS.

## 3.3 Falconn++: A locality-sensitive filtering approach

Falconn++ is an essential combination between the proposed LSF and the cross-polytope LSH of Falconn using sufficiently large $D$ random vectors. For each bucket, we simply filter out potential far away points if its projection value is smaller than $t = (1 - r^2/2)\sqrt{2\ln D}$. From Theorem 1, we have: $\ln(1/q_1) = \ln 2$ and $\ln(1/q_2) = \ln(\sqrt{2\pi}) + \ln\gamma + \gamma^2/2$, where $\gamma = \frac{cr(1-1/c^2)}{\sqrt{4-c^2r^2}} \cdot \sqrt{2\ln D}$. For a large $D$, we have $\ln(1/q_2) \approx \gamma^2/2 = \frac{(1-1/c^2)^2}{4/c^2r^2-1} \cdot \ln D$. The new exponent $\rho'$ of Falconn++ on answering $(c, r)$-NN is strictly smaller than the exponent $\rho$ of Falconn since

$$\rho' = \frac{\ln(1/q_1 p_1)}{\ln(1/q_2 p_2)} \approx \frac{\frac{\ln 2}{\ln D} + \frac{1}{4/r^2-1}}{\frac{(1-1/c^2)^2}{4/c^2r^2-1} + \frac{1}{4/c^2r^2-1}} \approx \frac{1}{1 + (1-1/c^2)^2} \cdot \frac{4/c^2r^2 - 1}{4/r^2 - 1} \leq \rho.$$

For a small $c$, $\rho \approx 1/c^2$ and $\rho' \approx 1/(2c^2 - 2 + 1/c^2)$. Figure 1b shows a clear gap between $\rho'$ compared to $\rho$ on various settings of $r$ and $c$.

**Discussion on the lower bounds.** The lower bound $\rho \geq 1/c^2 - o(1)$ [18, Theorem 4.1] for data-independent $(r, cr, p_1, p_2)$-sensitive LSH families on Euclidean space requires $p_2 = 1/n$ to yield an $\mathcal{O}\left(n^{1+\rho}\right)$ space and $\mathcal{O}\left(n^\rho\right)$ distance computations. Given the setting $p_2 q_2 = 1/n$, we have:

$$\left(\frac{(1-1/c^2)^2}{4/c^2r^2 - 1} + \frac{1}{4/c^2r^2 - 1}\right) \ln D = \ln n \Rightarrow D = n^{\frac{4/c^2r^2-1}{1+(1-1/c^2)^2}} = n^{\rho'(4/r^2-1)}.$$

To ensure Falconn++ has an $\mathcal{O}\left(dn^{\rho'}\right)$ query time, we need $D = \mathcal{O}\left(n^{\rho'}\right)$, and hence $r \geq \sqrt{2}$. We emphasize that the theoretical improvement of Falconn++ over the lower bound is only in a certain range of $r$. Note that this condition is also necessary for Falconn. Since nearest neighbors tend to be orthogonal, i.e. $r \approx \sqrt{2}$, in high dimensions, and distance computation often dominates hash evaluation, we can use a significantly large $D$ to boost the practical performance of Falconn variants.

**Connection to recent LSF frameworks.** Our proposed LSF mechanism combined with Falconn can be seen as a practical instance of the asymmetric LSF frameworks recently studied in theory [4, 8]. These frameworks use different filtering conditions on the data and query points to govern the space-time tradeoff of ANNS. In particular, given two thresholds $t_u, t_q$, over the random choice of $\mathbf{r}_i$, the

---
**Algorithm 1** Falconn++ with multi-probe indexing
---
1: **procedure** MULTI-PROBE INDEXING($\mathbf{X}, \alpha, D$ random vectors $\mathbf{r}_i, iProbes$)
2:     **for** each $\mathbf{x} \in \mathbf{X}$ **do**
3:         Compute top-$iProbes$ vectors $\mathbf{r}_i$ s.t. $|\mathbf{x}^\top \mathbf{r}_i|$ is largest.
4:         Add $\mathbf{x}$ into the $iProbes$ buckets corresponding to these $iProbes$ vectors.
5:     **end for**
6:     For each bucket of size $B$ corresponding to $\mathbf{r}_i$, keep top-$(\alpha B/iProbes)$ points with largest absolute projection values on $\mathbf{r}_i$.
7: **end procedure**
---

LSF frameworks study the probability where both $\mathbf{x}$ and $\mathbf{q}$ pass the filter corresponding to $\mathbf{r}_i$, i.e. $\mathbf{Pr}\left[\mathbf{x}^\top \mathbf{r}_i \geq t_u, \mathbf{q}^\top \mathbf{r}_i \geq t_q\right]$. Indeed, Falconn++ can be seen as a specific design of the asymmetric LSF framework where we utilize cross-polytope LSH to implement the filtering condition for queries and the asymptotic property of CEOs to implement the filtering condition for data points. We believe that our approach is much simpler in terms of theoretical analysis and practical implementation. Empirically, we observe that Falconn++ with practical add-ons provides higher search recall than the corresponding instantiation of the theoretical LSF frameworks.

### 3.4 Practical implementation and add-ons of Falconn++

In the previous section, we prove that there exists a threshold $t$ so that Falconn++ runs faster and uses less indexing space on answering $(c, r)$-NN than Falconn. Since the nearest neighbor distance is different from different queries in practice, it is difficult to find an optimal $t$ for all queries. We simplify the process of selecting $t$ by introducing a *scaling* factor $0 < \alpha < 1$ to control the fraction of number of points in a bucket. In particular, for any bucket of size $B$ corresponding to the random vector $\mathbf{r}_i$, we keep only the top-$(\alpha B)$ points $\mathbf{x}$ where $|\mathbf{x}^\top \mathbf{r}_i|$ is largest. Hence, the total number of points in one hash table is scaled to $\alpha n$.

Selecting an $\alpha$ fraction of points in a bucket is similar to selecting different values of $t$ on different density areas. In particular, we tend to use a larger $t$ for large buckets covering dense areas, and a smaller $t$ for small buckets covering sparse areas. This simple scheme can be seen as a *data-dependent* LSF without any training phase.

**Multi-probing for indexing: A new add-on.** We note that multi-probing is motivated by the observation that randomly perturbing $\mathbf{q}$ generates many perturbed objects that tend to be close to $\mathbf{q}$ and to be hashed into the same bucket with $\mathbf{q}$'s nearest neighbors [19]. This observation is also true in the reverse case where we perturb $\mathbf{q}$'s nearest neighbors to form new perturbed objects that tend to be hashed into the $\mathbf{q}$'s bucket. Hence, multi-probing can be used for indexing where we hash one point into multiple buckets. However, the downside of this approach is an increase of the table size from $n$ to $iProbes \cdot n$ points where $iProbes \geq 1$ is the number of indexing probes for one data point. Therefore, multi-probing for indexing has not been used in practice.

We note that the key feature of Falconn++ is the capacity to rank and filter out the points in any bucket based on their projection values to govern the space complexity. Given this mechanism, Falconn++ can utilize multi-probe indexing to improve the query performance without increasing the indexing space. Particularly, after executing $D$ random projections, we add $\mathbf{x}$ into the $iProbes$ buckets corresponding to the top-$iProbes$ random vector $\mathbf{r}_i$ closest or furthest to $\mathbf{x}$, as shown in Algorithm 1. Since we further scale each bucket by $iProbes$ times, the number of points in a table is still $\alpha n$. Empirically, a large $iProbes$ does not improve the performance of Falconn++ since we likely keep points closest to the random vector $\mathbf{r}_i$. When $\mathbf{q}$ is not close enough to $\mathbf{r}_i$, we will get all false positive candidates.

**Multi-probing for querying: A new criteria.** Among $D$ random vectors, the CEOs property asymptotically holds for the $2s_0$ closest and furthest random vectors to $\mathbf{q}$ for a constant $s_0 > 0$. Hence, Falconn++ can probe up to $2s_0$ buckets on each table. Unlike Falconn, we rank the probing sequences based on $|\mathbf{q}^\top \mathbf{r}_i|$ where $\mathbf{r}_i$ is the random vector corresponding to the probing bucket.

**New heuristics makes Falconn++ more efficient.** Besides the Structured Spinners [2] that exploit the fast Hadamard transform (FHT) to simulate Gaussian random projections in $\mathcal{O}(D \log D)$ time, we propose other heuristics to improve the performance of Falconn++.

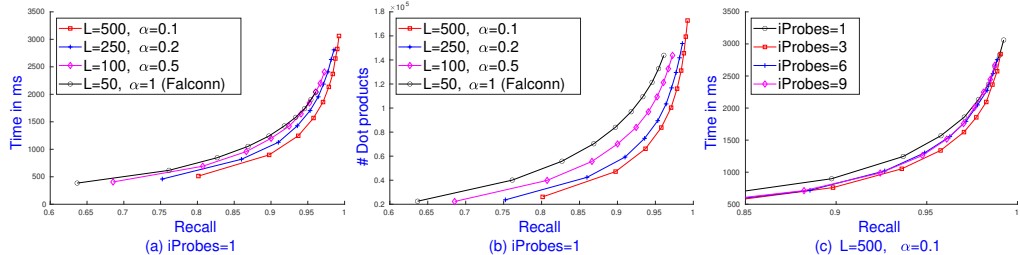

Figure 2: The recall-time tradeoffs of Falconn++ with various settings on Glove200.

**(1) Centering the data points.** We observe that when the data set distribution is skewed on a specific area of the sphere, Falconn variants do not perform well. This is because most data points will be hashed into a few buckets. To handle this issue, we center the data set before building the index. Denote by $\mathbf{c} = \sum_i \mathbf{x}_i / n$ the center of $\mathbf{X}$, we map $\mathbf{x}_i \mapsto \mathbf{x}_i' = \mathbf{x}_i - \mathbf{c}$. This mapping can diverge $\mathbf{X}$ to the whole sphere and hence increases the performance of Falconn variants. We note that after centering, $\arg\max_{\mathbf{x} \in \mathbf{X}} \mathbf{x}^\top \mathbf{q} = \arg\max_{\mathbf{x}' \in \mathbf{X}} \mathbf{x}'^\top \mathbf{q}$. Although $\|\mathbf{x}_i'\| \neq 1$, it does not affect the performance of Falconn++ since the CEOs property holds for the general inner product [20].

**(2) Limit scaling.** We observe that we do not need to scale small buckets since they might already contain high quality candidates. While we are scaling the bucket size on Line 6, Algorithm 1, we keep $max(k, \alpha B / iProbes)$ points. Note that this trick will increase space usage in practice. However, it achieves a better recall-speed tradeoff given the same space complexity configuration.

## 4 Experiment

We implement Falconn and Falconn++ in C++ and compile with `g++ -O3 -std=c++17 -fopenmp -march=native`. We conduct experiments on Ubuntu 20.04.4 with an AMD Ryzen Threadripper 3970X CPU 2.2 MHz, 128GB of RAM using 64 threads. We present empirical evaluations on top-$k$ NNS to verify our claims, including:

- Falconn++ with multi-probe indexing and querying provides higher search recall-speed tradeoffs than FalconnLib [1] given the same amount of indexing space.
- Falconn++ builds index faster and achieves competitive recall-speed tradeoffs compared to HNSW, [2] a recent efficient graph-based solution for ANNS on high search recall regimes.

We set $k = 20$ for all experiments. We use the average number of queries per second and Recall@20 to measure recall-speed tradeoffs. A better solution will give the recall-speed curve towards the top right corner. We conduct experiments on three popular data sets, including Glove200 ($n = 1,183,514, d = 200$), Glove300 ($n = 2,196,017, d = 300$), and NYTimes ($n = 290,000, d = 256$). We extract 1000 points to form the query set. All results are the average of 5 runs of the algorithms.

**A note on our implementation** (`https://github.com/NinhPham/FalconnLSF`). Our implementation uses `std::vector` to store hash tables and the Eigen library [3] for SIMD vectorization. We use `boost::dynamic_bitset` to remove duplicated candidates. For multi-threading, we add `#pragma omp parallel` directive on the `for` loop over 1000 queries. Compared to the released FalconnLib and HnswLib, our implementation is rather straightforward and lacks many optimized routines, such as prefetching instructions, optimized multi-threading primitives, and other hardware-specific optimizations. Given the same configuration, our Falconn's indexing time is 3 times slower than FalconnLib. Though Falconn++ still provides higher search-recall tradeoffs than these two highly optimized libraries, we believe there are rooms to engineer Falconn++ for further improvements.

**Falconn++ and FalconnLib parameter settings.** FalconnLib uses the number of bits to control the number of buckets in a hash table, which implicitly governs the number of concatenating hash functions $l$. In the experiment, our Falconn and Falconn++ use $l = 2$ and explicitly set the number of buckets via the number of random projections $D$. Since each hash function returns $2D$ different hash

---

[1]We use the latest version 1.3.1 from https://github.com/FALCONN-LIB/FALCONN

[2]We use the released version on Dec 28, 2021 from https://github.com/nmslib/hnswlib

[3]https://eigen.tuxfamily.org

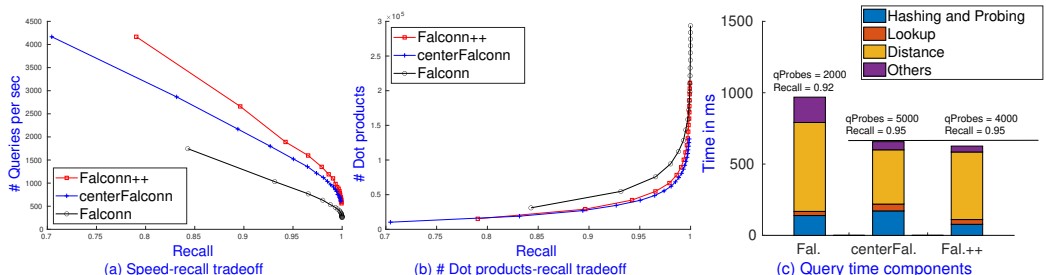

Figure 3: The performance of Falconn++ ($L = 500, \alpha = 0.1, iProbes = 3$) with new heuristics and its query cost components.

values, Falconn++ has $4D^2$ buckets on a hash table. For example, on Glove200 where $d = 200$, the setting $D = 256 = 2^8$ of Falconn++ leads to the suggested configuration of FalconnLib using 18 bits. Regarding the indexing space, both Falconn++ and Falconn use roughly the same memory footprint to contain the same number of points in a hash table.

## 4.1 An ablation study on Glove200

This subsection measures the performance of our implemented Falconn++ and Falconn. We show the advantage of functionalities of Falconn++, including multi-probe indexing and proposed heuristics for improving the recall-time tradeoff. We conduct the experiments on Glove200 using 64 threads. Both Falconn variants use $D = 256$. To have the same indexing space, Falconn uses $L$ hash tables and Falconn++ uses $L/\alpha$ where $\alpha$ is a scaling factor. Falconn++ with index probing $iProbes$ uses $L/(\alpha \cdot iProbes)$. Note that Falconn is Falconn++ with $iProbes = 1, \alpha = 1$.

**Falconn++ with multi-probing.** Figure 2 shows the recall-time tradeoffs of Falconn++ with various settings of $L$ and $\alpha$ such that the total number of points in all tables is $\alpha n L$. Falconn uses $qProbes = \{1000, \ldots, 10000\}$ and Falconn++ uses $qProbes/\alpha$ to have a similar number of candidates.

Without index probing, i.e. $iProbes = 1$, Figure 2a shows that Falconn++ has lower query time than Falconn given the same recall rate. Falconn++ achieves better recall-time tradeoffs when $\alpha$ is smaller. Figure 2b explains the superiority of Falconn++. Given $qProbes/\alpha$ probes, $\alpha = 0.1$ leads to the smallest candidate sizes for the same recall though Falconn++'s indexing time is increased by $1/\alpha$.

With the index probing, Falconn++ with $L = 500, \alpha = 0.1$ improves the performance with $iProbes = 3$ in Figure 2c. Larger $iProbes$ values are not helpful since Falconn++ tends to keep points closest or furthest to the random vectors only.

**Falconn++ with new heuristics.** Figure 3 shows the performance of Falconn++ with the centering and limit scaling heuristics. Given $L = 500, \alpha = 0.1, iProbes = 3$, the limit scaling that keeps $max(k, \alpha B/iProbes)$ points in any bucket increases the number of points in a table from $0.1n$ to $2.42n$ points. This is because many empty or small buckets tend to contain more points (up to $k = 20$) with index probing. However, such an increased number of points in a table pay off. Each bucket has sufficient high quality candidates and hence we do not need a large $qProbes$ to get high recalls. In this case, one hash table of Falconn++ can be seen as a "merged version" of 3 hash tables of Falconn.

Figure 3a shows the recall-speed comparison between Falconn++ with centering and limit scaling heuristics and Falconn variants given the same $2.42 \cdot nL$ points in the index. Though Falconn with the centering heuristic achieves higher tradeoffs than standard Falconn, both of them with $L = 1210$ tables are inferior to Falconn++. Figure 3b shows that the centering heuristic provides better candidates as Falconn++ and centering Falconn use a similar number of inner product computations to achieve the same recall. Figure 3c shows the query time breakdown components. Since Falconn++ uses less $L$ and $qProbes$, it has less hashing and probing time and hence less total query time compared to the centering Falconn given the same 95% recall ratio.

**Falconn++ parameter settings.** Falconn++ has several parameters, including $L, D, iProbes, \alpha, qProbes$. Since increasing $iProbes$ will degrade the performance (see Figure 2c), we need $iProbes$ large enough to ensure that each bucket has approximately $k$ candidates. Note that Falconn++ uses 2 LSH functions and hence has $4D^2$ buckets. With index probing, we expect the number

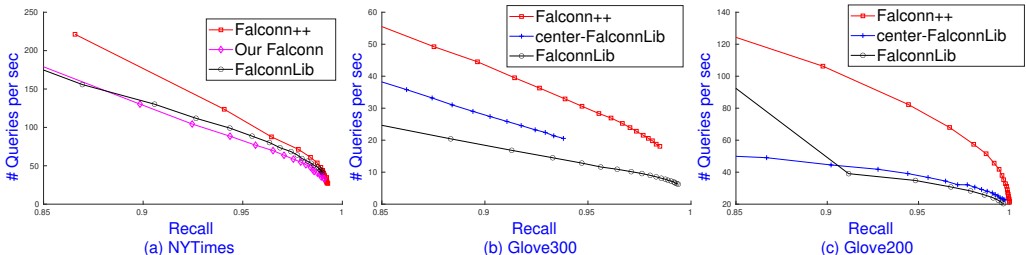

Figure 4: The recall-speed comparison between Falconn++ and FalconnLib on 3 data sets.

of points in a table, i.e. $n \cdot iProbes$, will be distributed uniformly into $4D^2$ buckets and each bucket has $n \cdot iProbes/4D^2$ points in expectation. We heuristically select $iProbes$ and $D$ such that $k \approx n \cdot iProbes/4D^2$. For instance, on Glove200: $n = 1M$, $D = 256$, $k = 20$, the choice of $iProbes = 3$, $D = 256$ leads to $1M \cdot 3/2^{18} = 2^4 = 16 < k = 20$ points in a bucket in expectation. $L$ and $\alpha$ depend on the available memory footprint, and $qProbes$ depends on the required recall ratio.

## 4.2 Comparison between Falconn++ and FalconnLib

This subsection compares our implemented Falconn++ with the released FalconnLib [2] given the same number of points in the index. Since FalconnLib does not support multi-threading for querying, Falconn++ uses 1 thread for a sake of comparison.

Falconn++ uses $L = 50$, $\alpha = 0.1$, $iProbes = 10$ on NYTimes, $L = 500$, $\alpha = 0.1$, $iProbes = 1$ on Glove300, and $L = 500$, $\alpha = 0.1$, $iProbes = 3$ on Glove200. $iProbes$ are set based on the above heuristic. For each data set, FalconnLib uses the corresponding scaled tables $L = \{200, 50, 1210\}$, respectively, to ensure the same space usage. We observe that FalconnLib needs $\{16, 18, 18\}$ bits for the corresponding data sets to yield the highest performance. For Falconn++, we set $D = \{128, 256, 256\}$ accordingly. For all approaches, we select $qProbes = \{1000, \ldots 20000\}$.

Figure 4 shows the recall-speed comparison between Falconn++ and FalconnLib on these 3 data sets. Since the center $\mathbf{c} \approx \mathbf{0}$ on NYTimes, the centering trick is not necessary. Instead, we plot our own implemented Falconn, which has lower recall-speed tradeoffs than FalconnLib. This again shows that our implementation is not well-optimized. Nevertheless, Falconn++ shows superior performance compared to FalconnLib on the 3 data sets. The significant improvement is from Glove300 and Glove200 where Falconn++ gains substantial advantages from better quality candidates and less number of query probes.

## 4.3 Comparison between Falconn++ and HNSW on high search recall regimes

This subsection compares Falconn++ with the released HnswLib [16], and Faiss-Hnsw [14], highly optimized libraries for ANNS using 64 threads. HNSW has two parameters to control the indexing space and time, including $ef\_index$ and $M$. We set $ef\_index = 200$ and use $M = \{1024, 512, 512\}$ for NYTimes, Glove300, and Glove200, respectively, to achieve high recalls with fast query time. Falconn++ uses $L = 500$, $\alpha = 0.01$, $iProbes = 10$ on NYTimes, $L = 900$, $\alpha = 0.01$, $iProbes = 1$ on Glove300, $L = 350$, $\alpha = 0.01$, $iProbes = 3$ on Glove200 to have the same space as HNSW.

Figure 5 shows the recall-speed tradeoffs of Falconn++ and HNSW libraries using 1 and 64 threads on NYTimes and Glove200. Falconn++ uses $qProbes = \{1000, \ldots, 20000\}$ and HNSW sets $ef\_query = \{100, \ldots, 2000\}$. Though Falconn++ is not competitive with HnswLib, it outperforms Faiss-Hnsw with 1 thread at the recall of 97%. With 64 threads, Falconn++ surpasses both HnswLib and Faiss-Hnsw from recall ratios of 0.97 and 0.94, respectively, on both data sets.

We observe that the speedup of Falconn++ comes from its simplicity and scalability on multi-threading CPUs. Table 1 shows that Falconn++ achieves higher threading scales than both HnswLib and Faiss-Hnsw when increasing the number of threads on NYTimes and Glove200. Both HNSW libraries suffer sufficiently suboptimal scales when the number of threads is larger than 16.

Figure 6 shows the recall-speed comparison between Falconn++, HnswLib, and brute force with the above parameter settings. Falconn++ provides higher recall-speed tradeoffs than HnswLib when

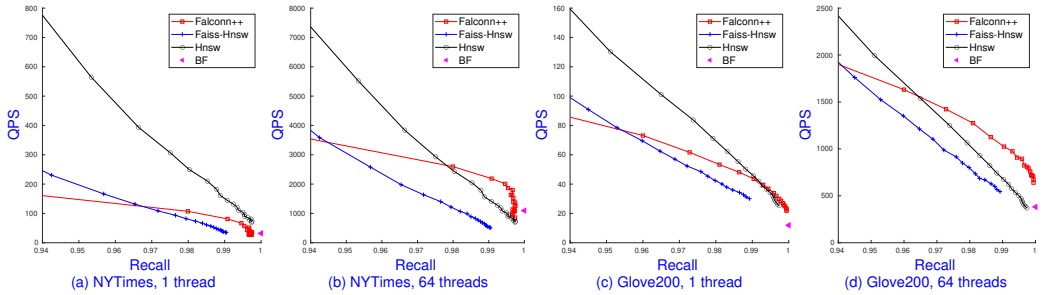

Figure 5: The recall-speed comparison between Falconn++, Faiss-Hnsw, and HnswLib on 1 thread and 64 threads.

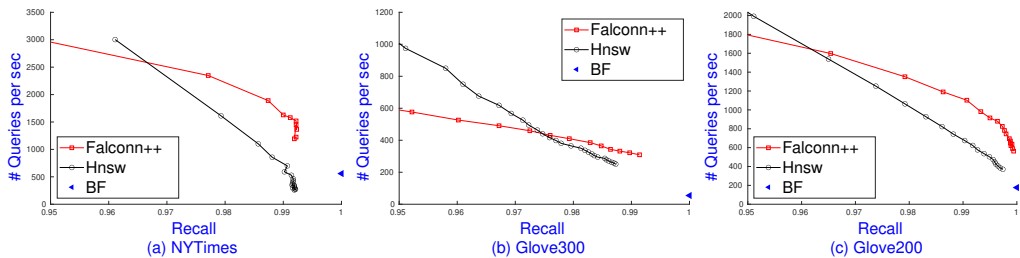

Figure 6: The recall-speed comparison between Falconn++ and HnswLib on 3 data sets.

the recall is at least 97% on the 3 data sets. Regarding the indexing space, they use nearly the same amount of RAM, as shown in Table 2. However, Falconn++ builds indexes at least 5 times faster than HNSW and hence can be better for streaming ANNS applications [23].

Due to the space limit, we present the comparison of recall-speed tradeoffs between Falconn++, the theoretical LSF frameworks, and other competitive ANNS solvers in the supplementary materials.

## 5 Conclusion

We present Falconn++, a practical locality-sensitive filtering scheme for ANNS on angular distance. Theoretically, Falconn++ asymptotically has lower query time complexity than Falconn, an optimal LSH scheme. Though our implementation of Falconn++ lacks many optimized routines, Falconn++ with several heuristic add-ons gives higher recall-speed tradeoffs than Falconn. It is also competitive with HNSW, an efficient representative of graph-based solutions on high search recall regimes.

Table 1: The average scale of Falconn++, Faiss-Hnsw, and HnswLib on a different number of threads on NYTimes. Similar suboptimal scales of Hnsw libraries are also observed on Glove200.

| Threads | 2 | 4 | 8 | 16 | 32 | 64 |
|---|---|---|---|---|---|---|
| HnswLib | 1.9 | 3.6 | 6.2 | 10.0 | 10.3 | 10.5 |
| Faiss-Hnsw | 2 | 3.7 | 6.9 | 10.9 | 14.6 | 15.1 |
| Falconn++ | 1.9 | 3.7 | 6.4 | 11.4 | 18.5 | 24.8 |

Table 2: Indexing space and time comparison between Falconn++ and HNSW on 3 data sets.

| Algorithms | NYTimes | | Glove300 | | Glove200 | |
|---|---|---|---|---|---|---|
| | Space | Time | Space | Time | Space | Time |
| Hnsw | 2.5 GB | 7.8 mins | 10.9 GB | 26.7 mins | 5.4 GB | 13.7 mins |
| Falconn++ | 2.7 GB | **0.6 mins** | 10.8 GB | **5.4 mins** | 5.3 GB | **1.1 mins** |

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
