# A supplementary for the paper
# Falconn++: A Locality-sensitive Filtering Approach for Approximate Nearest Neighbor Search

**Ninh Pham**
School of Computer Science
University of Auckland
ninh.pham@auckland.ac.nz

**Tao Liu**
School of Computer Science
University of Auckland
tliu137@aucklanduni.ac.nz

## 1 Proof of Theorem 1

**Theorem 1.** *Given sufficiently large $D$ random vectors and $c > 1$, the filtering mechanism described in the paper has the following properties:*

- *If $\|\mathbf{x} - \mathbf{q}\| \leq r$, then $\mathbf{Pr}\left[\mathbf{x} \text{ is not filtered}\right] \geq q_1 = 1/2$;*
- *If $\|\mathbf{y} - \mathbf{q}\| \geq cr$, then $\mathbf{Pr}\left[\mathbf{y} \text{ is not filtered}\right] \leq q_2 = \frac{1}{\sqrt{2\pi}\gamma} \exp(-\gamma^2/2) < q_1$ where*

  $\gamma = \frac{cr(1-1/c^2)}{\sqrt{4-c^2r^2}} \cdot \sqrt{2\ln D}.$

*Proof.* We first show the two properties for the case $\|\mathbf{x} - \mathbf{q}\| = r, \|\mathbf{y} - \mathbf{q}\| = cr$ by analyzing the tail of Gaussian random variables $X = \mathbf{x}^\top \mathbf{r}_1 \sim N(\mu_1, \sigma_1^2)$ and $Y = \mathbf{y}^\top \mathbf{r}_1 \sim N(\mu_2, \sigma_2^2)$, where

$$\mu_1 = \mathbf{x}^\top \mathbf{q} \sqrt{2\ln D} = (1 - r^2/2)\sqrt{2\ln D}, \sigma_1^2 = 1 - \left(1 - r^2/2\right)^2,$$

$$\mu_2 = \mathbf{y}^\top \mathbf{q} \sqrt{2\ln D} = (1 - c^2r^2/2)\sqrt{2\ln D}, \sigma_2^2 = 1 - \left(1 - c^2r^2/2\right)^2.$$

We use the classic tail bound of normal random variables. If $Z \sim N(0, 1)$, then for any $a > 0$,

$$\mathbf{Pr}\left[Z \geq a\right] \leq \frac{1}{a\sqrt{2\pi}} e^{-a^2/2}.$$

We define $\Delta\mu = \mu_1 - \mu_2 > 0$ and set the threshold $t = \mu_1 = (1 - r^2/2)\sqrt{2\ln D}$. Since $X \sim N(\mu_1, \sigma_1^2)$ and $t = \mu_1$, $\mathbf{Pr}\left[X \geq t\right] = 1/2 = q_1$. Applying the tail bound on $Y \sim N(\mu_2, \sigma_2^2)$,

$$\mathbf{Pr}\left[Y \geq t\right] = \mathbf{Pr}\left[\frac{Y - \mu_2}{\sigma_2} \geq \frac{\Delta\mu}{\sigma_2}\right] \leq \frac{1}{\sqrt{2\pi}(\Delta\mu)/\sigma_2} \exp\left(-\frac{(\Delta\mu)^2}{2\sigma_2^2}\right) = \frac{1}{\sqrt{2\pi}\gamma} \exp(-\gamma^2/2) = q_2,$$

where $\gamma = \Delta\mu/\sigma_2$. Since $\Delta\mu = \frac{c^2r^2}{2}(1 - \frac{1}{c^2})\sqrt{2\ln D}$ and $\sigma_2^2 = c^2r^2\left(1 - \frac{c^2r^2}{4}\right)$, we have $\gamma = \frac{cr(1-1/c^2)}{\sqrt{4-c^2r^2}} \cdot \sqrt{2\ln D}$.

Since $\Delta\mu/\sigma_2$ is monotonic with respect to $c$, further points has a higher probability of being discarded. Therefore, the second property holds for any far away point $\mathbf{y}$, i.e. $\|\mathbf{y} - \mathbf{q}\| \geq cr$. The first property holds for any close point $\mathbf{x}$, i.e. $\|\mathbf{x} - \mathbf{q}\| \leq r$, since their projection value onto $\mathbf{r}_1$ follows a Gaussian distribution with mean $\mu \geq \mu_1$. $\square$

## 2 Falconn++ vs. theoretical LSF frameworks

Figure 1 shows the recall-speed comparison between Falconn++ and recent theoretical LSF frameworks [2, 3]. All 3 data sets use $L = 100$, $\alpha = \{0.1, 0.5\}$, $iProbes = 1$, and the centering trick. We

do not apply the limit scaling trick to ensure that both Falconn++ and the theoretical LSF approaches share the same number of points in a table. We use $D = \{128, 256, 256\}$ for NYTimes, Glove200, and Glove300. With 2 LSH functions, each table of both approaches has the same $4D^2$ buckets.

Given $\alpha$, Falconn++ simply keeps $\alpha B$ points in a bucket of size $B$ whose absolute dot products to the corresponding vector $\mathbf{r}_i$ are the largest. To ensure that the table has $\alpha n$ points, theoretical LSF computes the global threshold $t_u$ such that it keeps $\mathbf{x}$ in the bucket corresponding to $\mathbf{r}_i$ with probability $\alpha/4D^2$. Since $\mathbf{x}^\top \mathbf{r}_i \sim N(0, 1)$, we use the `inverseCDF(.)` of a normal distribution to compute $t_u$ such that $\mathbf{Pr}\left[\mathbf{x}^\top \mathbf{r}_i \geq t_u\right]^2 = \alpha/4D^2$. Given this setting, the theoretical LSF with pre-computed $t_u$ has a similar number of indexed points as Falconn++.

Figure 1 shows superior performance of Falconn++ compared to the theoretical LSF with $\alpha = \{0.1, 0.5\}$ on 3 data sets. Note that NYTimes has the center vector $\mathbf{c} = \mathbf{0}$, hence does not need centering. Figure 1(b) and (c) show that Falconn++ with centering trick can even improve the performance on Glove200 and Glove300, whereas theoretical LSF significantly decreases the performance with centering trick. This is because the LSF mechanism of Falconn++ can work on the general inner product (after centering the data points) while the theoretical LSF mechanism works on a unit sphere.

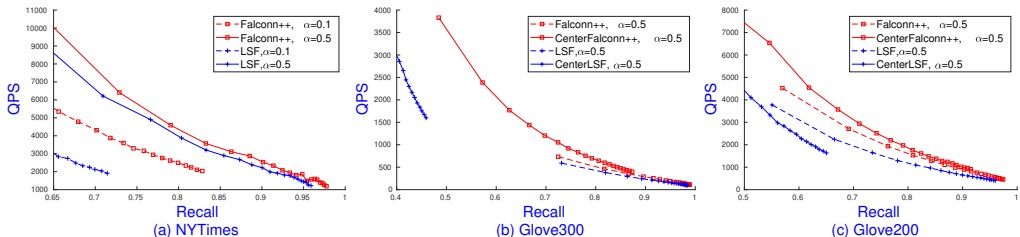

Figure 1: The recall-speed comparison between Falconn++ and theoretical LSF frameworks.

## 3   A heuristic to select parameter values for Falconn++

This section will present a heuristic to select parameters of Falconn++, including number of random vectors $D$, number of tables $L$, scale factor $\alpha$, and number of indexing probing $iProbes$.

Since Falconn++ uses 2 LSH functions, the number of buckets is $4D^2$. We apply the limit scaling trick to keep $max(k, \alpha B/iProbes)$ points in any bucket. Since we expect to see approximately $k$ near neighbors in a bucket, this trick prevents scaling small buckets that might contain top-$k$ nearest neighbors. When applying $iProbes$, we expect the number of points in a table, i.e. $n \cdot iProbes$, will be distributed equally into $4D^2$ buckets. Hence, each bucket has $n \cdot iProbes/4D^2$ points in expectation.

We note that we would not want to use large $iProbes$ since the bucket will tend to keep the points closest to the random vector $\mathbf{r}_i$, and therefore degrades the performance. Falconn++ with a large $iProbes$ works similarly to the theoretical LSF framework [2, 3] which keeps the point $\mathbf{x}$ in the bucket corresponding to $\mathbf{r}_i$ such that $\mathbf{x}^\top \mathbf{r}_i \geq t_u$ for a given threshold $t_u$. LSF frameworks need to use a large $t_u$ so that a bucket will contain a small number of points to ensure the querying performance. Figure 1 shows Falconn++ with a *local* threshold $t$ adaptive to the data in each bucket, outperforms the theoretical LSF frameworks that use a *global* $t_u$ for all buckets.

The heuristic idea is that we select $iProbes$ and $D$ such that the bucket size has roughly $k$ points in expectation by setting $k \approx n \cdot iProbes/4D^2$. For instance, on Glove200: $n = 1M$, $D = 256$, $k = 20$, each table has $4D^2 = 2^{18}$ buckets. The setting $iProbes = 3, D = 256$ leads to $1M \cdot 3/2^{18} = 2^4 = 16 < k = 20$ points in a bucket in expectation.

Falconn++ needs a sufficiently large $D$ to maintain the LSF property. Since we deal with high dimensional data set with large $d$, $D \approx 2^{\lceil \log_2 d \rceil}$ is sufficient. Falconn++ with larger values of $D$ and $iProbes$ requires larger memory footprint but achieves higher recall-speed tradeoffs, as can be seen in Figure 2.

On NYTimes with $n = 300K$, we set $L = 500$, $D = \{128, 256\}$, $iProbes = \{10, 40\}$. On Glove200 with $n = 1M$, we set $L = 350$, $D = \{256, 512\}$, $iProbes = \{3, 10\}$. On Glove300 with

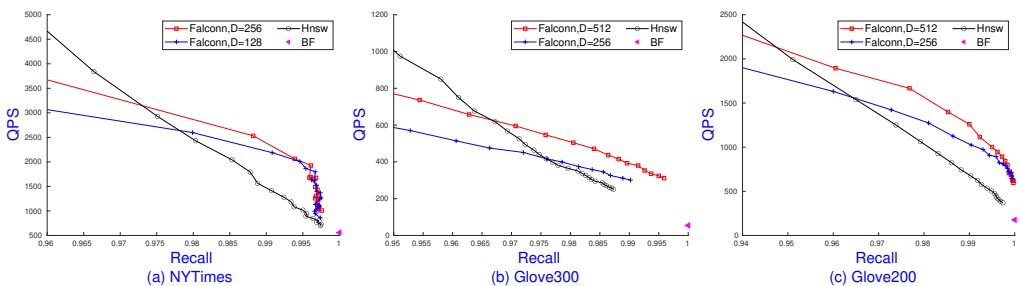

Figure 2: The recall-speed comparison between Falconn++ and HNSW with various $D$.

$n = 2M$, we set $L = 900$, $D = \{256, 512\}$, $iProbes = \{1, 4\}$. On all 3 data sets, the first setting of $D$ and $iProbes$ lead to similar memory footprints of HNSW. The second setting increases the indexing space to approximately 4 times since we double $D$.

Regarding $\alpha$, given the scaling limit trick, we set $\alpha = 0.01$ to reduce large buckets without affecting the performance. We observe that $\alpha = \{0.01, \ldots, 0.1\}$ gives the best performance without dramatically changing the indexing size.

## 4 Comparison between Falconn++ and HnswLib on different top-$k$ values on Glove200

Figure 3 shows the recall-speed tradeoffs between Falconn++ and HNSW on several values of $k = \{1, 5, 10, \ldots, 100\}$ on Glove200 with $L = 350$, $D = 256$, $iProbes = 3$, $\alpha = 0.01$. Since we apply the limit scaling trick to keep $max(k, \alpha B/iProbes)$ points in any bucket, Falconn++ does not work well on small $k = \{1, 5, 10\}$, compared to HNSW in Figure 3(a). This is due to the fact that many high quality candidates in a bucket are filtered away with $\alpha = 0.01$. However, Falconn++ can beat HNSW for larger $k$, i.e. at recall ratio of 0.95 for $k \geq 60$ and at recall ratio of 0.96 for $20 \leq k \leq 50$ in Figure 3(b) and (c).

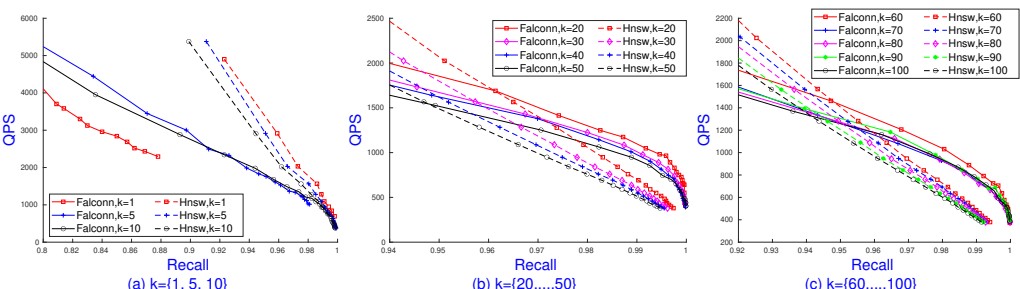

Figure 3: The recall-speed comparison between Falconn++ and Hnsw on different $k$ with the scaling limit $max(k, \alpha B/iProbes)$.

To deal with small $k$, we set the limit scaling to $max(\kappa, \alpha B/iProbes)$ where $\kappa = 20$ to maintain enough high quality candidates in a bucket without affecting indexing time and space (see in Table 1 for $k = \{1, 5, 10\}$). Figure 4 shows that Falconn++ with the setting of $max(20, \alpha B/iProbes)$ is competitive with HNSW at recall ratio of 0.93 for $k = 1$, and recall ratio of 0.96 for $k = \{5, 10\}$ given the same indexing size.

## 5 Comparison between Falconn++ and other state-of-the-art ANNS solvers

This section will give a comprehensive comparison between Falconn++ with other state-of-the-art ANNS solvers, including ScaNN [4], Faiss [5], and coCEOs [7] on high search recall regimes on three real-world data sets, including NYTimes, Glove200, and Glove300. The detailed data sets are

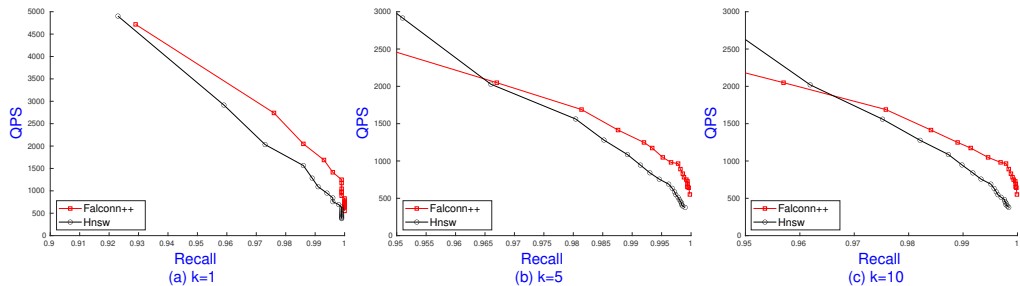

Figure 4: The recall-speed comparison between Falconn++ and Hnsw on different $k$ with the scaling limit $max(20, \alpha B/iProbes)$.

on Table 3. For ScaNN, we use the latest version 1.2.6 released on 29 April, 2022. [1] For FAISS, we use the latest version Faiss-CPU 1.7.2 released on 11 January, 2022. [2] For coCEOs, we use the latest released source code. [3] We note that ScaNN does not support multi-threading while Falconn++, FAISS and coCEOs do though their thread-scaling is not perfect.

**Parameter settings of Falconn++.** Since Falconn++ uses 2 concatenating cross-polytope LSH functions and $D$ random projections, there are $4D^2$ number of buckets in a hash table. Since we focus on $k = 20$, we set $D = 2^b$ where $b \approx \lceil \log_2 (n/k) \rceil /2$ to expect that each bucket has roughly $k$ points. Hence, we use $D = \{128, 256, 256\}$ for NYTimes, Glove300, and Glove200, respectively. This setting corresponds to $\{2^{16}, 2^{18}, 2^{18}\}$ buckets in a hash table on three data sets. Note that the setting of $D$ is proportional to the size of the data sets. The hash function is evaluated in $\mathcal{O}(D \log D)$ time, so it does not dominate the query time. Furthermore, these values of $D$ are large enough to ensure the asymptotic CEOs property.

We note that Falconn++ with the heuristics of centering the data and limit scaling make the bucket size smaller and more balancing. We observe that different small values of $\alpha$ does not change the size of Falconn++ index. Hence, to maximize the performance of Falconn++, we set $\alpha = 0.01$. For the

---

[1] https://github.com/google-research/google-research/tree/master/scann
[2] https://github.com/facebookresearch/faiss
[3] https://github.com/NinhPham/MIPS

Table 1: Hnsw takes **13.7 mins** to build 5.4GB indexing space. Falconn++ takes **1.1 mins** and needs different memory footprints dependent on $k$. For $k \leq 10$, we use $max(20, \alpha B/iProbes)$. For $k \geq 20$, we use $max(k, \alpha B/iProbes)$.

| $k$ | 1 | 5 | 10 | 20 | 30 | 40 | 50 | $60 - 100$ |
|---|---|---|---|---|---|---|---|---|
| Falconn++ | 5.3GB | 5.3GB | 5.3GB | 5.3GB | 5.8GB | 6.0GB | 6.1GB | 6.2GB |

Table 2: Indexing space and time comparison between Falconn++ and HNSW on 3 data sets.

| Algorithms | NYTimes | | Glove300 | | Glove200 | |
|---|---|---|---|---|---|---|
| | Space | Time | Space | Time | Space | Time |
| Hnsw | 2.5 GB | 7.8 mins | 10.9 GB | 26.7 mins | 5.4 GB | 13.7 mins |
| Falconn++ | 2.7 GB | **0.6 mins** | 10.8 GB | **5.4 mins** | 5.3 GB | **1.1 mins** |

Table 3: Data sets.

| | NYTimes | Glove300 | Glove200 |
|---|---|---|---|
| $n$ | 290,000 | 2,196,017 | 1,183,514 |
| $d$ | 256 | 300 | 200 |

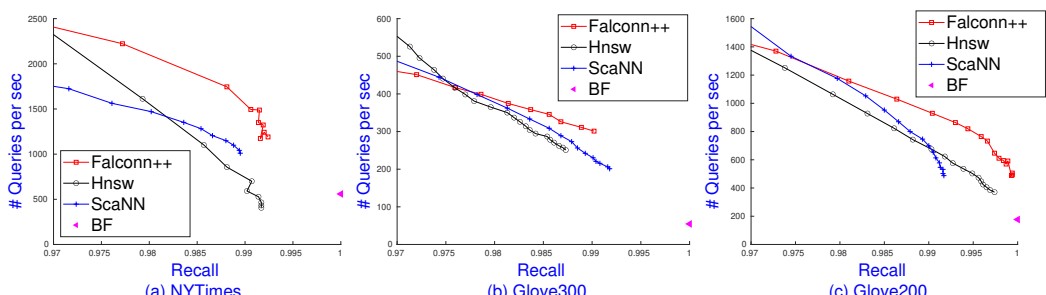

Figure 5: The recall-speed comparison between Falconn++, HNSW, and ScaNN on 3 data sets.

sake of comparison, we first select optimal parameter settings for HNSW [6] to achieve high search recall ratios given a reasonable query time. Based on the size of HNSW's index, we tune the number of hash tables $L$ for Falconn++ to ensure that Falconn++ shares a similar indexing size with HNSW but builds significantly faster, as can be seen in Table 2. In particular, we use $L = 500, iProbes = 10$ for NYTimes, $L = 900, iProbes = 1$ for Glove300, and $L = 350, iProbes = 3$ for Glove200. Since the characteristics of the data sets are different, it uses different values of $iProbes$.

**Parameter settings of HNSW.** We first fix $ef\_index = 200$ and increase $M$ from 32 to 1024 to get the best recall-speed tradeoff. Then, we choose $M = \{1024, 512, 512\}$ for NYTimes, Glove300, and Glove200, respectively. We observe that changing $ef\_index$ while building the index does not improve the recall-speed tradeoff. We vary $ef\_query = \{100, \ldots, 2000\}$ to get the recall ratios and running time.

**Parameter settings of ScaNN.** We used the suggested parameter provided in ScaNN's GitHub. We use *all* points to train ScaNN model with $num\_leaves = 1000$ and $score\_ah(2, anisotropic\_quantization\_threshold = 0.2)$. For querying, we use $pre\_reorder\_num\_neighbors = 500$ and vary $leaves\_to\_search \in \{50, \ldots, 1000\}$ to get the recall ratios and running time.

**Parameter settings of FAISS.** We compare with Faisee.IndexIVFPQ and set the sub-quantizers $m = d, nlist = 1000$, and 8 bits for each centroid. We again use *all* points to train FAISS. We observe that $m < d$ or increasing $nlist$ returns lower recall-speed tradeoffs. We vary $probe \in \{50, \ldots, 1000\}$ to get the recall ratios and running time.

**Parameter settings of coCEOs.** We use $D = 1024$ and $SamplingSize = n, s_0 = 20$, and vary the number of candidates from 10,000 to 100,000 to get the recall ratios and running time.

**Comparison of recall-speed tradeoffs.** Figure 5 shows that Falconn++, though lacking many important optimized routines, achieves higher recall-speed tradeoffs when $recall > 0.97$ compared to both HNSW and ScaNN on all three data sets. We emphasize that the speed of ScaNN and HNSW comes from several optimized routines, including pre-fetching instructions, SIMD in-register lookup tables [1] for faster distance computation, and optimized multi-threading primitives. Compared to HNSW and ScaNN, both FalconnLib and Falconn++ simply use the Eigen library to support SIMD vectorization for computing inner products.

Figure 6 and 7 shows that Falconn++ achieves higher recall-speed tradeoffs than both FAISS and coCEOs over a wide range of recall ratios. Since coCEOs is designed for maximum inner product search, its performance is inferior to other ANNS solvers for angular distance.