# OpenReview forum: "Falconn++: A Locality-sensitive Filtering Approach for Approximate Nearest Neighbor Search"
_NeurIPS.cc/2022/Conference — NeurIPS 2022 Accept_

### Official Review · Reviewer_y5Sd · 2022-07-12

**Rating:** 6
**Confidence:** 4
**Soundness:** 3 good
**Presentation:** 2 fair
**Contribution:** 3 good

**Summary:**

The paper proposes an interesting extension of the special LSH scheme for angular distance (and cosine similarity) nearest neighbor search. Specifically it proposes a filtering mechanism and proves that it leads to asymptotically better retrieval times. Empirically, the approach is competitive with the graph-based retrieval algorithm HNSW for high recall values.

Nearest neighbor search is a backbone of many recommender and retrieval systems. Clearly, it is important to be able to do this task well. There has been historically a competition of several approaches including LSH, randomized trees, quantization approaches, and graph-based retrieval methods with sometimes comparable outcomes. It has been a useful competition, because different methods have different strengths and applicability. In my opinion, this is an important line of research.

The paper proposes an interesting trick to index-time filtering of unpromising data points. This is possible, because the inner product between a bucket "pivot" and a query is quite close to the *SCALED* inner product between the query and data points in a bucket (BTW it's nice to point out that scaling arises from the fact that data/query are normalized but "pivot"/sampled vectors are not). Authors further show this property can be exploited both theoretically and practically to improve upon FALCON.

In terms of theory, they show it for a c-approximate nearest neighbor search. For practical purposes, one has to resort to theory-inspired heuristics. I think it's totally fine, but the paper should be clear about it.

Empirically, the method improves upon the original FALCON and claims to be comparable to HNSW, but only in high-recall zone when the number of threads are large. Although this was not in the original version of the paper, additional experiments revealed that HNSW implementations seems to scale poorly with the number of threads, at least for the specific architecture used in the paper.

So, potentially HNSW can work better, but we don't know this for sure. This is unfortunate, but resolving this issue may require a separate paper. I personally think that the most important baseline in this work is FALCON. Plus, if the approach is more scalable (with the number of threads), this is certainly a plus.


**Questions:**

**Some detailed comments:**

25-27, please give a clearer definition of a decision version of the problem.

34 citation is needed.

45-46 not only that but old-style random-projection LSH often performs no better than a VP-tree modified for approximate search. They are also typically outperformed by random-projection trees. If I remember correctly authors of FLANN have some comparison points.

123-124 how is it different from the original FALCON ordering of buckets in line 115? In lines 234-237 you remove the absolute value, is this actually the difference?

126 a significantly large NUMBER of vectors ??? Same issue in line 169.

127 Eq (3) can possibly be accurate for x == q, because we get zero variance. I guess it should have some error term diminishing with D.

Also lines 128-129 We can see that, among D random vectors, if r1 is closest (or furthest) to q, then the projection values of *ALL* points in X preserves their inner product order expectation -> I feel this is a very inaccurate description. First of all, the equation doesn't say anything about expectation of *ALL* points. It does say that the distance is preserved with a high probability P for a given pair of points. If you have three points, the full order preservation is going to be only P^2, etc... for a large enough set it's going to be close to zero IMHO. However, I believe the expected number of preserved distances should be high.

135-143 This is confusing. If the filtering heuristic is used at indexing time, it is not related to multi-probing. It should improve performance of a regular single-probe LSH as well, because unlikely candidates are going to be pruned.

173-174  If this is index-time filtering why does it depend on the query q? I think I understand what you are talking about, but some explanation won't harm.

241 The centering approach is not new and it was in fact used in FALCON when FALCON was still present in ANN-BENCHMARK. Please, see, e.g., the code here:

https://github.com/erikbern/ann-benchmarks/blob/406574ef659338accb6a926a097099c1691ef8f3/ann_benchmarks/algorithms/falconn.py#L29




**Limitations:**

1. negative societal impact discussion isn't really applicable.
2. authors discuss limitations of their implementation as well as limitations of approximate nearest neighbor search in general.

**Strengths And Weaknesses:**


1. The paper is a bit messy and many explanations are confusing. Although with some effort the paper is understandable, but the presentation can be improved substantially.
2. The centering trick is NOT NEW and it was used in FALCON when it was incorporated into ANN BENCHMARKS. Please see a reference below.

Generally, math makes sense to me, but I did not check very fine details. In particular, I have to trust the derivation of the Eq. (3) from prior work, but I haven't checked this derivation myself.

https://github.com/erikbern/ann-benchmarks/blob/406574ef659338accb6a926a097099c1691ef8f3/ann_benchmarks/algorithms/falconn.py#L29

---

> ### Author Response · Authors · 2022-08-02
> **The claim that Falconn++ is competitive to Hnsw is correct.**
>
> Thank you for the review. We address your mentioned weakness and questions below.
>
> For the raising issue of flawed numbers in experiments, we have cross-checked our experiment settings, and all are correct. Your claim is based on the assumption that all implemented software/libraries should achieve the same scale while using the same number of used threads. Unfortunately, this is NOT correct. Even a library (e.g. Faiss) can return different scale on different parameter settings given the same number of used threads.
>
> We re-run the experiments on NYTimes on different numbers of threads $\{1, 2, 4, 8, 16, 32, 64\}$ with our BF, Falconn++, Hnsw, and Faiss. The average speedup factors of the used methods for the corresponding number of threads are as follows.
> - BF:             {2, 3.6, 5.5, 12.5, 24.4, 34}
> - Hnsw:         {2, 3.5, 6.2, 9.4, 10.9, 10.6} on different parameter settings, e.g. $m = 1024, m = 512$ and ef-query in {100, ... 2000}
> - Falconn++: {2, 4, 6.5, 11.8, 20.5, 34.6}
> - Faiss:         {2, 4, 8, 16, 32, 42} for $nlist = 10000, probes = 1000$
> - Faiss:         {2, 4, 7.4, 14.3, 18.5, 19.7} for $nlist = 1000, probes = 1000$
> - Faiss:         {2, 4, 8, 14.5, 27, 44} for $nlist = 1000, probes = 100$
>
> On Glove200, Hnsw provides similar scales as NYTimes while Faiss gives 38x and Falconn++ gives 26x with 64 threads.
> Therefore, we confirm that our empirical results on 64 threads are correct.
>
> W1: We agree that chasing SOTA on a benchmark is a great motivation. However, it is not sufficient to evaluate the contribution of a new algorithm based on the comparison of its straightforward implementation (Line 266 - 274) with highly-optimized libraries that have been contributed by many researchers and developers in several years.
> Especially, making ANNS run efficiently requires significant engineering efforts. For example, Scann runs faster due to the use of SIMD in-register lookup tables (see https://medium.com/@kumon/similarity-search-scann-and-4-bit-pq-ab98766b32bd, and https://github.com/facebookresearch/faiss/wiki/Fast-accumulation-of-PQ-and-AQ-codes-(FastScan)).
> Indeed, we argue that our current implementation of Falconn++ with multi-threading is competitive with Hnsw due to the fact that Falconn++ is simple to implement, has significantly less overhead on multi-threading, and builds index significantly faster given the same indexing space.
>
> We believe that after releasing the source code on Github and engaging the community contribution, Falconn++ will become one of strong baselines of ANNS, especially on streaming ANNS applications.
>
> W3: We would like to justify your understanding of the filtering mechanism: The inner product is also preserved for points in nearby buckets as well (see Lemma 2 and Line 132 - 137), and both queries and data do not need to be normalized (Lemma 2 holds for the non-normalized cases in [19]). This explains why indexing probing is helpful; and why Falconn++ and FalconnLib use different centering tricks. Falconn applies centering trick to both data and queries while Falconn++ applies only on data (Line 241 - 247 for more details). Since applying centering trick to queries leads to a new set of queries, the performance of Falconn will be degraded.
>
> Q25-27. This is a boolean variant of (c, r)-NN where you return YES if there is a point within r, and return NO if no point within cr.
>
> Q123-124. Regarding implementation, it is not different. If a far way random vector $r_1$ is selected, we use $-r_1$ as the hash value and the projection is still $|r_1 \cdot q|$. If we use 1 LSH function to construct a table, Falconn++ and FalconnLib build the same query probing sequence. However, they are different since we use 2 LSH functions e.g. $r \in R, s \in S$. Falconn++ considers both projection values of $q$, e.g. $|r_1 \cdot q| + |s_1 \cdot q|$ as the ranking score, while FalconnLib considers $max(|r_1 \cdot q|, |s_1 \cdot q|)$ as the ranking score. Building the probing sequences among $L$ tables leads to different results. We observe that Falconn++ gives higher recall with the new probing criteria.
>
> Q126. We need $D$ to be infinitive to use the sharp bound of normal distribution. In practice, we observe that $D > 100$ is sufficient.
>
> Q128-129. It is a typo, ... their inner product order "in" expectation.
>
> Q135-143. I do not understand your question. The filtering mechanism is used while constructing the index to reduce any bucket of size $B$ to $\alpha \cdot B$, and it can also improve the single-probe LSH. Since a table now is scaled by $\alpha$, we can use $L/\alpha$ tables to have the same indexing space but get higher quality candidates. The problem with single-probe LSH is that we need a significantly large number of tables to have good recall. Hence multi-probing queries are utilized to reduce the space usage.
>
> Q173-174. The threshold depends on $r, c, D$, not $q$. We just apply it on all buckets. We need $h(q)$ since it matches the filtering mechanism that requires $h(q)$.

---

> > ### Comment · Reviewer_y5Sd · 2022-08-04
> > **scaling issues should be considered separately**
> >
> > Thank you for clarifications. It's correct that scaling is never perfect, but I am yet to see the case when HNSW scaling is substantially suboptimal as observed in your case. Did you try different HNSW implementations? Furthermore, it will be IMHO important to disentangle thread-scaling issue and algorithm effectiveness. Even if HNSWLIB has some scaling issues, you are presenting comparison results for the most favorable case. Do your results generalize to Intel CPUs with 16 cores, etc... ?

---

> > > ### Author Response · Authors · 2022-08-04
> > > **... but not for potentially simple and scalable algorithms**
> > >
> > > We observed that Hnsw gives a max scale around 10 though the number of used threads are more than 10. However, Faiss and our implementation do provide better scale even for 16 threads. Therefore, we conjecture that there should be a sufficient engineering effort to re-implement Hnsw to utilize multi-threading. In this case, the new efficient multi-threading implementation might not be as fast as the current single-thread implementation if measured on a single thread. We have not tested different Hnsw implementations, and we did not test on other machines.
> > >
> > > We note that our implementation is quite straightforward and lacks many optimized routines compared to Hnsw. For multi-threading, we only add 1 line: $#pragma omp parallel for$ and nothing else, while Faiss has a separate class to control the threads. Regarding engineering efforts to make the algorithm run fast, Falconn++ is minimal among compared approaches. We do not present a comparison for the most favourable case, it comes from the simplicity and scalability of Falconn++. We will explain in detail where the speedup comes from in the manuscript.
> > >
> > > If we disentangle the thread-scaling issue and algorithm effectiveness, we are putting down potentially simple and scalable algorithms.  Since multi-threading architecture is popular on many modern CPU, and we note that Faiss and Hnsw use multi-threading to build indexes, why not execute the querying process in multi-threads with just 1 line of code?

---

> > > > ### Author Response · Authors · 2022-08-09
> > > > **More comparison with Faiss-Hnsw added in supplementary. Still, Faiss-Hnsw scale up to 15x with 64 threads.**
> > > >
> > > > We have added Section 4 in the supplementary that includes a comparison between Falconn++ and Faiss-Hnsw and HnswLib. We both run experiments on 1 thread and 64 threads. Both Faiss-Hnsw and HnswLib suffer from scaling issues when using more than 16 threads, while Falconn++ does not.

---

> > > > > ### Comment · Reviewer_y5Sd · 2022-08-09
> > > > > **...**
> > > > >
> > > > > Thank you for running additional experiments. I do agree we need to run the methods in the multi-threading mode. However, it would be nice to understand if
> > > > > 1. HNSW poor scaling is related to inherent limitations of the algorithm (i.e., it is a poor fit for the modern architectures) or
> > > > > 2. It is a deficiency of a specific implementation (or CLASS of implementations)
> > > > > 3. Possibly exacerbated on a SPECIFIC architecture.
> > > > > This is why I do find this finding very interesting, but it should come with a proper disclaimer with a reference to additional experiments in the appendix.
> > > > > Please, note that I don't require you to rewrite the paper and to rerun all experiments, I ask you to document this part better.

---

### Official Review · Reviewer_spwu · 2022-07-15

**Rating:** 6
**Confidence:** 4
**Soundness:** 4 excellent
**Presentation:** 4 excellent
**Contribution:** 4 excellent

**Summary:**

This paper investigated a fundamental problem of approximate nearest neighbor search on angular distance. The authors developed a practical LSF approach called Falconn++ that combines the LSF property with the LSH to achieve a lower query time complexity beyond the vanilla LSH schemes such as Falconn. Extensive experiments validated the superior performance of Falconn++.


**Questions:**

Q1. Followed by W1, can they conduct more experiments for parameter settings? Or provide more theoretical (or empirical) analysis for the impact of the parameters.

Q2. Followed by W2, can they compare Falconn++ with FalconnLib and HNSW for different values of k (e.g., from 1 to 100)?

Q3. Can they discuss the possibility of extending the pattern of LSF + LSH to support the ANNS on other similarity/distance measures?

**Ethics Review Area:**

["I don’t know"]

**Limitations:**

Yes.

**Strengths And Weaknesses:**

Strengths

S1. The pattern of LSF + LSH is very interesting to me. Even though the two key components (i.e., CEO and Falconn) of Falconn++ come from existing work, they discover a promising way to combine them together and provide a comprehensive theoretical analysis.

S2. They conducted a systematic comparison of Falconn++ with Falconn and HNSW. Extensive results justify their theoretical analysis and validate the superior performance of Falconn++.

S3. The paper is well organized and easy to follow.

Weaknesses

W1. The proposed Falconn++ contains many parameters, such as D, L, $\alpha$, iProbes, qProbes. In the experiments (Subsections 4.2 and 4.3), they config different values of L, $\alpha$, iProbes for Falconn++ when comparing with different methods even for the same datasets. Thus, it might be more comprehensive and beneficial for practitioners if they could make a discussion for the parameter settings.

W2. The experiments might be more convincing if they could study the sensitivity of k for Falconn++.

---

> ### Author Response · Authors · 2022-08-02
> **We can provide a heuristic for selecting the suitable parameters and provide a graph for sensitivity of k**
>
> Thank you for the review. We address your mentioned weakness and questions below.
>
> W1 and Q1: The parameter $\alpha$ controls the # points in a bucket and hence the indexing space. However, when using the limit scaling trick that keeps $max(k, \alpha \cdot B \cdot iProbes)$ points in a bucket and for a large $D$, $\alpha$ is only sensitive for large buckets.
> This is because most of the buckets tend to be filled by just a few points. For instance, on Glove200: $n = 1M$, $iProbes = 3$, $D = 256$, $k = 20$, each table uses 2 LSH functions, hence has $4D^2 = 2^{18}$ buckets. In expectation, there will be roughly $1M \cdot 3 / 2^{18} = 2^4 = 16 < k = 20$ points in a bucket. Therefore, we can use a small $\alpha = 0.01$ to reduce the size of large buckets without affecting the performance.
>
> Given $\alpha = 0.01$, setting $iProbes$ depends on $n, D, k$. We expect $k \approx n \cdot iProbes / 4D^2$. Indeed, we follow this approach to select the parameters for our experiments.
>
> Larger $L$ provides a higher speed-recall tradeoff but costs more indexing space. We try to set $L$ so that Falconn++ shares the same Hnsw's indexing space and show that Falconn++ builds index much faster.
>
> Indeed, we can improve the speed-recall tradeoff by increasing $D$ and $iProbes$. For example, Glove200 with $D = 512, iProbes = 10$ can beat Hnsw from recall = 0.95, instead of 0.97 when $D = 256, iProbes = 3$.
>
> W2: We observe the same improvement of Falconn++ over Hnsw when $k = 10$. We can provide an experiment for different $k = \{1, ..., 100}$. However, given the current implementation of Falconn++, we guess Falconn++ can beat Hnsw for larger $k$ ,e.g. $k \geq 10$, but not for smaller $k$, e.g. $k = 1$.
>
> Q3: We can extend Falconn++ to support L2 and inner product. The key idea is the property of CEOs hold for a generic inner product. Indeed, the proposed centering trick (Line 241 - 247) transforms the inner product search in a unit sphere to the general inner product search where points and queries do not necessarily have unit norms. For L2, by an asymmetric mapping, we can transform L2 into the inner product.

---

> > ### Comment · Reviewer_spwu · 2022-08-06
> > **Extra questions for the response**
> >
> > Thank you for your explanation about the extension of Falconn++. The answer is clear to me.
> >
> > Nonetheless, your response to W1 and W2 is not convincing to me. For example, for W1, why do you consider $k \approx n \cdot iProbes / 4D^2$? Can you provide more insight on the formula? For W2, you claim Falconn++ can beat HNSW when $k \geq 10$ but inferior for smaller $k$. Can you explain why this happens? Does it because Falconn++ filters too many data points in the buckets (or for other reasons)?
> >
> > In addition, NeurIPS 2022 allows authors to revise their papers and supplementary with more results and clarifications to address the issues raised by the reviewers, but I cannot find any new results for your response. Could you provide any results to support your claims about parameter setting and the comparison with HNSW about the sensitivity to $k$?

---

> > > ### Author Response · Authors · 2022-08-09
> > > **Section 1 and 2 are added in the supplementary to address your request**
> > >
> > > W1: We would not want to use large $iProbes$ since it will degrade Falconn++ performance. In particular, large $iProbes$ will make points in a bucket closer to the random vectors. In this case, the performance of Falconn++ is similar to the theoretical LSF framework which keep the point $x$ based on $x \cdot r_i \geq t_u$. We also expect approximately $k$ candidates on each bucket for the top-$k$ NNS. Therefore, we set $iProbes$ just large enough so that the number of points $iProbes \cdot n$ distribute into $4D^2$ buckets, and each bucket receives  $k$ points in expectation.
> > >
> > > W2: If we use the scaling limit trick that keeps $max(k, \alpha B / iProbes)$, then Falconn++ with $\alpha = 0.01$ is inferior for small $k$. This is because good candidates are filtered out. However, when $k \geq 20$, buckets of Falconn++ contain sufficient candidates so its performance is better. To improve Falconn++ for small $k \leq 10$, we change the limit to  $max(20, \alpha B / iProbes)$. Hence, Falconn++ can beat Hnsw on a high recall regime.
> > >
> > > W1 & W2: We provide the empirical comparison between Falconn++ and Hnsw to verify our claims on the rebuttal version of the supplementary.

---

### Official Review · Reviewer_1qZS · 2022-07-26

**Rating:** 6
**Confidence:** 3
**Soundness:** 3 good
**Presentation:** 4 excellent
**Contribution:** 3 good

**Summary:**

The author proposed a locality-sensitive filtering approach with an asymptotically lower query time algorithm than Falconn and competitive to efficient indices like HNSW. Falconn++ keeps only the top-($\alpha B/iProbes$) points (closest to the hyperplane $r_i$) in each bucket. The closeness is defined by the threshold on the inner product distance $t = (1 − r 2 /2) 2 log D$, from the $r_i$. A limit scaling strategy is added for more balanced buckets and higher query-recall performance.

**Questions:**

Questions:
1) Where are the significant query time improvements (empirical) coming from compared to Falconnlib? Is it due to filtered far-away points in buckets or the heuristics?
2) I am not able to locate the metric. Everywhere it is mentioned as a recall. Is it Recallk@k where k is 20 or something else?
3) How does Falconn++ compares with Fast cross polytope LSH*?

*Kennedy, Christopher, and Rachel Ward. "Fast cross-polytope locality-sensitive hashing." arXiv preprint arXiv:1602.06922 (2016).

**Limitations:**

The authors have addressed their work's limitations and potential negative societal impact.

**Strengths And Weaknesses:**

Strengths

1) Theoretical analysis is adequate and conveys the proposed approach's correctness.

2) The paper is well written. The preliminary ideas and the state of art methods are thoroughly discussed.

3) Experiments with SOTA ANN approaches: The work has compared Falconn++ with Falconnlib, HNSW, FAISS, ScANN, and coCEOs over Million scale datasets. Falconn++ beats these state-of-the-art approaches in the high recall region.

4) New heuristics have shown improvement over the proposed LSF approach.

Weakness

1) It is great to see the comment on the comparison with other LSFs[4][8].  It will be good to see the experimental comparison as well for thorough evaluation. (citations from the paper)

2) No performance evaluation in lower recall regime- especially for HNSW and ScANN; however, comparison with coCEOs covers almost the entire spectrum. ScANN is really good in the high recall region, and it is great to see that Falconn++ is very close to it and even beats it on smaller-scale datasets. However, it will be helpful to analyze how Falconn++ performs against them in the mid-recall region (0.5-0.8), as the near neighbors in mid-recall are also useful for many practical search systems. Same argument for FAISS and HNSW.

---

> ### Author Response · Authors · 2022-08-02
> **W1 has already implemented in the submitted code, W2 is difficult given the current unoptimized code, but possible in future**
>
> Thank you for the review. We address your mentioned weakness and questions below.
>
> W1: We implemented the theoretical LSF [4, 8] in the submitted code where we selected a GLOBAL threshold $t$ to ensure that the whole table is scaled by $0 < \alpha < 1$. Since $r \cdot x \sim N(0, 1)$, we select $t$ s.t. $Pr[r \cdot x \geq t] = \alpha$ by using the inverse of the standard normal cdf. Empirically, this approach does not provide good recall since it tends to keep the points $x$ with large $r \cdot x$. Its performance is similar to Falconn++ with a significantly large $iProbes$, since it tends to keep points closer to the random vector.
>
> Since Falconn++ scales each bucket by $\alpha$, it selects different thresholds $t$ on different buckets and provides better candidate quality, especially for queries in sparse areas where points are often located far away from the random vector. On theoretical LSF, these points have small dot products, and tend to be filtered out from the index (See Line 213 - 216 in the paper). We will add an experiment to demonstrate this finding.
>
> W2: For the mid-recall (0.5 - 0.8), it seems that our current implementation of Falconn++ cannot to beat Hnsw and Faiss since it lacks many optimized routines.
> Comparing to FalconnLib, our Falconn implementation is 3 times slower on building the index.
> Comparing to industry SOTA libraries which are contributed by many researchers and developers, significant engineering efforts are necessary, and we believe there are rooms for further engineering Falconn++ (see Line 266 - 274).
> While our current implementation Falconn++ cannot beat highly-optimized Hnsw and Faiss libaries, Falconn++ has many advantages, e.g. simpler to run in parallel, and faster for updating index for streaming ANNS where data-dependent approaches (Hnsw, Faiss) cannot provide. We will release the source code on GitHub and engage the contribution from the community to make Falconn++ as one of baselines for ANNS.
>
> Falconn++ uses multi-threading for both indexing and querying, hence it is better than Scann that uses multi-threading for building index, and single thread for querying.
> Our rebuttal discussion with Reviewer #3 will highlight the key engineering trick that makes Scann faster than other product-quantization based approaches, e.g. Faiss.
>
> Q1: From Fig 2b, you can see that given the same # candidates (# dot product computations), Falconn++ with smaller $\alpha$ achieves higher recall.
> This observation holds when compared against FalconnLib.
>
> When applying centering trick on Glove200, the buckets tend to be more balanced and the limit scaling tricks are applied on a few dense buckets.
> In this case, $iProbes$ plays the role since it increases the size of a table to roughly $iProbes$ times by hashing $x$ $iProbes$ times.
> Compared to FalconnLib, Falconn++ saves $iProbes$ times from evaluating the hash values and preparing probing sequences with the same indexing size and the same # candidates.
> Additionally, the criteria for constructing query probing sequences of Falconn++ do provide higher search recall than FalconnLib.
>
> Q2: Yes, it is Recall@20 since we use k = 20 for the whole experiment.
>
> Q3: Before answering this question, we would like to point out a small mistake on estimating $\rho'$ of Falconn++ in Line 190 - 191. Indeed, it should be
> $\frac{1/q_1p_1}{1/q_2p_2} \approx \frac{\log{2} / \log{D} + \frac{1}{4/r^2 - 1}}{(1 + (1 - 1/c^2)^2)\frac{1}{4/c^2r^2 - 1}} \approx \frac{1}{1 + (1 - 1/c^2)^2}\frac{4/c^2r^2 - 1}{4/r^2 - 1} \approx \frac{1}{2c^2 - 2 + 1/c^2}$
>
> The $\rho'$ of Falconn++ matches the initial locality-sensitive filtering (LSF) work* in theory for a large $c$.
> Falconn++ can be seen as a practical variant of LSF that matches the asymptotical performance with a simpler proof given the assumption that $D$ tends to be infinity.
>
> *Becker et al. New directions in nearest neighbor searching with applications to lattice sieving, https://dl.acm.org/doi/10.5555/2884435.2884437
>
> The work fast cross-polytope LSH provides the theory for the used heuristic trick that utilizes random rotations (via Fast Hadamard Transform) to simulate Gaussian random projections.
> This trick will reduce the cost of hash evaluation from $O(dD)$ to $O(D\log{D})$ (see Line 238 - 240).
> While this work only considers the running time of the hash evaluation, Falconn++ studies and improves the search efficiency with an asymptotically smaller $\rho$ parameter.

---

> > ### Author Response · Authors · 2022-08-09
> > **New comparisons between Falconn++ and other theoretical LSF[4][8] are added in Section 3 in the supplement**
> >
> > For W1, Section 3 which compares Falconn++ and other LSFs is added. Falconn++ outperforms other LSFs on all 3 data sets.

---

> > ### Comment · Reviewer_1qZS · 2022-08-09
> > **Thanks for the response**
> >
> > Thanks to the authors for the response. W1 is clear. Q1 and Q3 are clear. Q2: Thanks. Adding the formal definition in the paper about all the metrics in use will help. W2: I agree with the authors, and yes, Faiss and HNSW are well optimized. So it seems there is no way to quantify and analyze how Falconn++ performs against these baselines in mid recall region? The argument seems inconclusive to me.
> > Maybe the number of operations (e.g., flops) and index size comparison with Faiss, Scann, and other baselines can also give the idea of this.
> >
> > Related to this, is there a specific reason why you chose Faiss-hnsw and not Faiss-IVF for the comparison?

---

> > > ### Author Response · Authors · 2022-08-10
> > > **Both Faiss-IVFPQ and Faiss-HNSW are used**
> > >
> > > Both Faiss-IVFPQ (Section 6) and Faiss-HNSW (Section 4) are used in the supplement. Another reviewer raises the request to compare Falconn++ with other Hnsw implementations. Hence, we just add Faiss-Hnsw into the supplementary (rebuttal version).
> > >
> > > W2: With the current implementation, we agree that it's difficult for Falconn++ (data-independent approach) to beat libraries like Faiss, Scann, or Hnsw (data-dependent approaches) on mid-recall regions.

---

> > > > ### Comment · Reviewer_y5Sd · 2022-08-10
> > > > **...**
> > > >
> > > > I would argue that it's not necessary to beat HNSW right away, this can be another paper. Improving FALCON is sufficient as long as benchmarks are accurate.

---

> > > > > ### Author Response · Authors · 2022-08-10
> > > > > **...**
> > > > >
> > > > > Thanks. Since Falconn does not support multi-threading, we use 1 thread to measure the query time of Falconn++ and Falconn. Both use the same indexing space, so the benchmark should be fine.

---

### Meta-Review · Area_Chair_zVUq · 2022-09-06

**Recommendation:** Accept
**Confidence:** Certain

**Metareview:**

The paper provide a good and exciting improvement over LSH based widely used Falcon Library. All the reviewers found the contribution worthy of publication.

**Award:**

No

---

### Decision · Program_Chairs · 2022-09-14

Accept